# Interplay between Thickness, Defects, Optical Properties, and Photoconductivity at the Centimeter Scale in Layered GaS

**DOI:** 10.3390/nano12030465

**Published:** 2022-01-28

**Authors:** Stefano Dicorato, Yael Gutiérrez, Maria M. Giangregorio, Fabio Palumbo, Giuseppe V. Bianco, Maria Losurdo

**Affiliations:** Institute of Nanotechnology, CNR-NANOTEC, c/o Dipartimento di Chimica, Università di Bari, via Orabona 4, 70126 Bari, Italy; stefano.dicorato@nanotec.cnr.it (S.D.); yael.gutierrezvela@nanotec.cnr.it (Y.G.); michelaria.giangregorio@nanotec.cnr.it (M.M.G.); fabio.palumbo@cnr.it (F.P.); giuseppevalerio.bianco@cnr.it (G.V.B.)

**Keywords:** gallium monosulfide, layered chalcogenide, UV-photoresponsivity, ellipsometry

## Abstract

From the group-III monochalcogenide (MX, M  =  Ga, In; X  =  S, Se, Te) layered semiconductors, gallium monosulfide, GaS, has emerged as a promising material for electronics, optoelectronics, and catalysis applications. In this work, GaS samples of various thicknesses in the range from 38 to 1665 nm have been obtained by mechanical exfoliation to study the interplay between structural, morphological, optical, and photoresponsivity properties as a function of thickness. This interplay has been established by analyzing the structure through Raman spectroscopy and X-ray diffraction, the morphology through scanning electron microscopy and atomic force microscopy, the density and optical properties through spectroscopic ellipsometry, and the photoresponsivity through current–voltage measurements under UV light. This work shows that photoresponsivity increases with increases in GaS thickness, resulting in a UV photoresponsivity of 1.5·10^−4^ AW^−1^ stable over several *on/off* cycles.

## 1. Introduction

Layered group-III monochalcogenide semiconductors MX (M = Ga and In, X = S, Se, and Te) have emerged as potential candidates for photocatalytic [1,2] and optoelectronic applications [3,4] due to their stability, tunable bandgap, and high carrier mobility. Among those, gallium monosulfide, GaS, is a layered van der Waals (vdW) semiconductor with a hexagonal lattice, where each layer consists of S–Ga–Ga–S atoms covalently bonded and six-membered Ga_3_S_3_ rings. Each S-chalcogen atom is sp^3^ hybridized, with three of the sp^3^ orbitals forming Ga–S bonds; the remaining S-chalcogenide orbital is occupied by lone-pair electrons forming vdW interactions between layers, as shown in Figure 1. GaS holds a special place in the family of layered monochalcogenides because of its wide energy bandgap, which extends up to the UV region. Specifically, GaS is a wide, indirect bandgap semiconductor with experimental bandgap values reported in the range from 2.34 to 2.62 eV depending on the deposition technique [5,6,7,8,9,10], whereas values of 3.02 eV [11] and 3.33 eV [12] of the indirect bandgap have been reported for bilayer and monolayer GaS, respectively.

A few layers of GaS can be deposited by different methods, such as chemical vapor deposition (CVD) [13,14], transport reaction [5,9], pulsed laser deposition [10,15], and mechanical exfoliation [3,16].

The recent interest in GaS stems from its exploitation in photocatalysis, electrochemical hydrogen production by water splitting [17], energy storage [18], nonlinear optics [19], gas sensing [20], and ultraviolet selective photodetectors [21,22]. Among various approaches to photodetectors, the low-dimensional semiconductor-based photodetectors are very attractive because of the relatively easy fabrication by exfoliation; nevertheless, subsequent mechanical exfoliation that reduces the thickness results in ripples, wrinkles, cracks, and the non-homogeneous thickness of flakes.

In this paper, we discuss the dependence on the thickness of the interplay between the structural and optical properties and the photoresponsivity of GaS samples obtained by mechanical exfoliation. The investigated samples have a thickness greater than 30 nm (which corresponds to approximately 30 GaS layers); this has been verified using optical microscopy, which shows that continuous flakes on a centimeter scale can be obtained for this thickness. The mechanically exfoliated GaS layers have been transferred onto glass substrates and characterized structurally using Raman spectroscopy and X-ray diffraction (XRD), morphologically by scanning electron microscopy (SEM) and by atomic force microscopy (AFM), and optically using spectroscopic ellipsometry. We show that photoresponsivity decreases with the decrease in GaS thickness, which is correlated to the density, polarizability, and light absorption of the GaS layers. Finally, we investigate the applicability of the exfoliated GaS samples to UV-light detection, demonstrating good cyclability and *on/o**ff* current ratios in the range from 10^4^ to 10^5^.

## 2. Materials and Methods

Grown by the Bridgman method, *c*-axis (0001)-oriented hexagonal Gallium (II) Sulfide (β-GaS) crystals were purchased by 2D Semiconductor (nominal purity > 99.99%). Mechanical exfoliation by the thermal tape release method was performed on the as-received crystals, with subsequent transferring onto Gorilla glass substrates of GaS flakes. By subsequent exfoliations, samples with a thickness in the range from 38 to 1665 nm were obtained.

Raman spectroscopy was performed using a LabRAM HR Horiba spectrometer (HORIBA Ltd., Palaiseau, France), with a 532 nm wavelength continuous excitation laser source. Measurements were carried out on several points of the samples in the 50–1000 cm^−1^ range at room temperature.

After exfoliation, the crystalline quality was checked by X-ray diffraction (XRD) in thin film geometry. The measurements were performed with an Ultima IV diffractometer (Rigaku Corp., Akishima, Japan), equipped with parallel beam optics and a thin film attachment, using Cu Kα radiation (λ = 1.5405 Å), operated at 30 mA and 40 kV, over the 2θ range from 5 to 70°, at a scanning rate of 1°/min, with a step width of 0.02°.

Scanning Electron Microscopy (SEM) was performed with a Zeiss Supra 40 FEG SEM (Zeiss, Oberkochen, Germany) equipped with a Gemini field emission gun operated at an extraction voltage of 3 kV and a 30 µm aperture.

XPS measurements were carried out using a Scanning XPS Microprobe (PHI 5000 Versa Probe II, Physical Electronics, MN, USA) equipped with a monochromatic Al Kα x-ray source (1486.6 eV) with a spot size of 200 µm. Survey (0–1200 eV) and high-resolution spectra (C 1s, O 1s, S2p, S2s, Ga2p3, Ga3d, and valence band region) were recorded in FAT mode at a pass energy of 117.40 and 29.35 eV, respectively. Spectra were acquired at a take-off angle of 45° with respect to the sample surface. Surface charging was compensated using a dual- beam charge neutralization system, the hydrocarbon component of C1s spectrum was used as an internal standard for charging correction, and it was fixed at 285 eV.

Spectroscopic Ellipsometry (HORIBA Ltd., Palaiseau, France) was used to measure the room temperature spectra of the refractive index and the extinction coefficient in the 0.75–6.5 eV photon energy range with 0.05 eV resolution. Ellipsometric spectra at the incident angle of 70° were herein represented, although variable angle spectra in the range from 55 to 75° were acquired to check possible anisotropy of the samples, with the crystal structure being hexagonal. Indeed, because of the mosaicity of the various flakes and defects from the exfoliation, the isotropy assumption worked well. The experimental spectra of the complex pseudodielectric function, <ε> = <ε_1_> + i<ε_2_>, were fitted by an ambient air/GaS + voids/glass model. The glass substrate was measured before transferring the GaS. The GaS dielectric function from a single crystal [23] was combined to voids representing the defects according to the Bruggeman effective medium approximation (BEMA) [24] under isotropic fitting assumption.

After the deposition of silver (Ag)-paste electrodes (~50 nm thick), placed 0.5 cm apart on top of the GaS layers, electrical characterization was performed using a Keithley617 Programmable Electrometer (Keithley Instruments, OH, USA), scanning voltages from −10 to +10 V. Photoresponse was investigated using an AM1.5 spectrum halogen 100 mW cm^−2^ lamp and a 405 nm laser (250 mW cm^−2^) as visible and UV sources, respectively.

## 3. Results

### 3.1. Structural and Optical Characterization

Figure 1 shows the centimeter scale exfoliated GaS layers of different thicknesses transferred onto glass, with the peculiar yellow color (due to the bandgap of ≈2.50 eV) vanishing with the decrease in thickness caused by a decrease in light absorption. The change in color is due to the change in thickness and not in the indirect bandgap, which stays at 2.50 eV in the investigated range of thickness; specifically, by DFT calculations and experimental ellipsometric measurements, it was discovered that, for a number of layers higher than five (approximately 5 nm), the bandgap value has a bulk value. The morphological features, obtained by scanning electron microscopy (SEM) and atomic force microscopy (AFM) for the samples of various thicknesses, clearly indicate that the layered structure of the samples with structural defects, i.e., cracks, wrinkles, ripples, and flake boundaries, increase with the number of exfoliations and, hence, with the decrease in the thickness. The thickness values reported in Figure 1a were obtained by the fitting of ellipsometric spectra (see the Discussion section). The reported errors result from the ellipsometric analysis on a spot size of 2 mm × 4 mm. These thicknesses are compared to those in Figure 1e obtained by the AFM line profiles, where the standard deviation was calculated on ten line profiles in various positions per sample. AFM topographies also show that, at the layer level, the surface is very smooth and clean, ruling out residuals from the tape transfer.

The crystalline quality and structure of the exfoliated samples were assessed by Raman scattering and X-ray diffraction. The XRD diagram of the thin 38 nm exfoliated sample in Figure 2a shows prominent peaks corresponding to the GaS (002) and GaS (004) reflections of the *c*-axis oriented *2H* structure of the β-GaS hexagonal phase, with lattice constants of *a* = *b* = 0.3585 nm and *c* = 1.5500 nm, and the (006) and (0010) reflections visible in the inset. The aforementioned are visible on all of the exfoliated samples. By increasing the number of exfoliations to reduce the thickness, additional peaks corresponding to the GaS (105) and GaS (107) reflections appear due to different planes exposed by the flakes’ edges.

Figure 2b shows the Raman spectra of the samples with different thicknesses. From the irreducible representation of the D^4^_6h_ point group, GaS presents phonon modes A^1^_1g_ at 187 ± 1 cm^−1^, A^2^_1g_ at 359 ± 1 cm^−1^, and E^1^_2g_ at 294 ± 1 cm^−1^.

The intensity of all Raman modes decreases with the decrease in thickness; the E^1^_2g_ mode becomes barely visible in the few-layer regime, which is consistent with the trends already reported in the literature [25,26]. The ratio A^1^_1g_/A^2^_1g_ ~ 1, typical of the single crystals, is preserved for all exfoliated samples, indicating that the layers’ stacking is unaffected by the exfoliation process. Furthermore, there is no significant shift in the frequency of both the A^1^_1g_ and A^2^_1g_ Raman modes; consistent with previous reports on GaS, wavenumber shifts are approximately 1 cm^−1^, which is within the error bar of peak position indicated. This is different from what is typically observed in other bidimensional chalcogenides, such as MoS_2_ [27], where the position of Raman modes can be used to identify the number of layers as the wavenumber of the MoS_2_ A_1g_ mode decreases whereas that of E_2g_ increases with the decrease in thickness. The softening of the in-plane E_2g_ mode (of about 4 cm^−1^ from bulk to monolayer) in MoS_2_ with the increase in thickness is due to the decrease in the long-range Coulombic interaction between Mo atoms. In contrast, the out-of-plane A_1g_ mode wavenumber increase (approximately 5 cm^−1^ from monolayer to bulk) with the increase in thickness is due to restoring forces on the S-atoms due to interlayer interactions. Therefore, the negligible shift for GaS indicates a negligible interaction between the layers and negligible stress/strain in the GaS samples stemming from the exfoliation. On the contrary, as shown in Figure 2c, the full width at half maximum (FWHM) of the peaks increases with the decrease in thickness because of the larger density of defects induced by the exfoliation, which is consistent with the SEM images in Figure 1b. The increased density of wrinkles and edges also affects the oxidation of the samples as indicated by the XPS data in Figure 2e–f. Specifically, the O/Ga atomic percentage ratio increases with the exfoliation runs and with the decrease in GaS thickness. Correspondingly, the fitting of the Ga3d and S2p photoelectron core levels shows a higher contribution of Ga–O and defective sulfur at the high binding energy side for the thin GaS sample, which is consistent with the assumption that GaS oxidation occurs preferentially at the edges and wrinkles.

The optical properties, as determined by spectroscopic ellipsometry, are shown in Figure 3. The spectra of the real, ⟨ε_1_⟩, and imaginary, ⟨ε_2_⟩, parts of the pseudodielectric function, ⟨ε⟩ = ⟨ε_1_⟩ + *i*⟨ε_2_⟩, show an interference system from which the sample thickness has been derived through a fitting routine using the model sketched in the inset. For the GaS layer, we used a BEMA mixture of the GaS dielectric function from [23] and *voids*, which represent defects; the % volume fraction of *voids* increases with the decrease in thickness consistently with the increase in defects and cracks seen via SEM. The spectra of the imaginary part of pseudodielectric function, ⟨ε**_2_**⟩, show two main peaks at 3.95 eV and at 5.45 eV corresponding to interband critical points, with the former attributed to the transitions between the highest valence band and the lowest conduction band in the neighborhood of the high symmetry points K and H in reciprocal space where the bands run parallel and the latter assigned to electronic transition along the Γ–M and A–L paths between the deeper valence and the conduction band. The fundamental indirect energy gap E_0_ is at 2.50 eV, and it does not change in the investigated range of thickness. Interestingly, the amplitude of the critical points decreases with the decrease in thickness. Good agreement was observed between the ellipsometric thickness and the AFM line profiles. Figure 3c,d show, respectively, the effective refractive index (which includes the voids representing the defects) and effective extinction coefficient at 3.06 eV (i.e., 405 nm corresponding to the UV light wavelength used during the photoresponsivity measurements, as shown below) as a function of the thickness of the GaS exfoliated samples. From these plots, it can be inferred that, by decreasing the GaS thickness, the effective density and light absorption of samples decreases because of the finite dimension of the flakes. Furthermore, the shadowed yellow region indicates that two regimes of thickness can be identified, namely for thicknesses below 1000 nm, where the effective density and light absorption increases with thickness, whereas a plateau is reached above 1000 nm; however, this is lower than the bulk GaS crystal because of the defects introduced by the exfoliation.

### 3.2. Electrical Characterization

Figure 4a,b show the I–V characteristics of the thick (1665 nm) and thin (38 nm) GaS samples under dark, visible, and UV (405 nm) light irradiation. The dark current is in the range 10^−13^ A–10^−15^ A for the thick and thin samples, which is important for good detectivity in photodetectors. In the dark, the resistivity of the thick sample is 4.8 × 10^7^ Ω·cm, in agreement with the literature [26,28,29]; it decreases to 8.2·10^4^ Ω·cm and 1.1·10^3^ Ω·cm upon illumination with visible and UV light, respectively. For the thin sample, the resistivity decreases from the dark value of 1.3·10^7^ Ω·cm to 7.4·10^6^ Ω·cm and 5.9·10^6^ Ω·cm upon visible and UV light irradiation, respectively. This is due to the lower light absorption indicated by the lower refractive index, i.e., polarizability and extinction coefficient (as shown in Figure 3). For samples thicker than 1000 nm, the photocurrent, which increases by 4-5 orders of magnitude, is linear with the applied voltage in the −10 V to +10 V range (see Figure 4a), whereas a slight thermal displacement is seen for the thin sample (see Figure 4b).

In order to gain further insights into the effect of the exfoliation process and thickness on the photoresponsivity, Figure 4c–d show that the photocurrent increases with the incident optical power. The data plotted in a log–log scale show a linear trend consistently with the equation:(1)$$I_{ph}=A\cdot P^{\alpha },$$
where *A* is a quantity related to the photoresponsivity and *α* is a dimensionless parameter containing information on the traps present in the system, α = 1 is expected in the case of a trap-free photodetector and decreases in the presence of traps. From Figure 4c, α = 0.61 and α = 0.77 are derived for the thick GaS, under visible and UV light, respectively. Conversely, thin GaS (see Figure 4d) has α = 0.16 and α = 0.34 for visible and UV light, respectively, which is consistent with the higher density of defects shown by SEM in Figure 1; a higher degree of oxidation, shown in Figure 2 by XPS; and higher voids percentage in the ellipsometry model, shown in Figure 3. A non-unity exponent of 0 < *α* < 1 in Equation (1) is often found in the low-dimensional photodetectors as a result of the complex process of carrier generation, trapping, and recombination within low-dimensional semiconductors with large surface-area-to-volume ratios, which is due to the long-lived states from the surface and interface traps [30]. As reported by Szałajko et al. [30], one of the possible processes involved in the nonlinear behavior of the photoresponsivity of GaS is the surface recombination, causing a decrease in the GaS velocity of carriers with the increase in illumination intensity.

The lower value of *α* obtained under visible light irradiation indicates the contributions of trap states below and within the bandgap originated by structural defects, while UV irradiation (405 nm is above the energy gap of GaS) mainly involves band-to-band states. Trap states also influence the dynamics of the GaS photodetectors [31]. According to Equation (1), the photoresponsivity (*R*), defined as
(2)$$R=\frac{I_{ph}}{P_{opt}},$$
decreases with the increase in the power density of light, following $R∼\;P^{1-\alpha }$ as shown in Figure 4c–d, where *R* is a parameter evaluating the sensitivity of the photoresponsive system, *I_ph_* is the photocurrent, and *P_opt_* is the light power. If the photocurrent linearly depends on the light intensity, *R* is a constant; conversely, in Figure 4c–d, *R* decreases with the increase in incident power, which is typical for most photodetectors that use exciton separation. As the excitation power increases, more photoinduced electron–hole pairs are generated, and their interactions cause a loss of energy and efficiency during charge extraction [32]. Under high illumination intensities, the density of available states is reduced, resulting in a saturation of the photoresponsivity.

The photoresponsivity under UV light is 1.5·10^−4^ A·W^−1^ when measured at 405 nm on centimeter-scale GaS compared with 2.3·10^−4^ A·W^−1^ when measured at 490 nm by Hu et al. [16], as well as with photoresponsivity of other bidimensional materials, as summarized in Table 1. The switching *on/off* of the photocurrent, i.e., the dynamic of the photo-to-dark current ratio (*PDCR*), or *on/off* ratio, defined as the ratio between the photocurrent *I_ph_* and the dark current *I_off_*,
(3)$$PDCR=\frac{I_{ph}}{I_{off}}=\frac{I_{on}-I_{off}\;}{I_{off}},$$ is shown in Figure 4e–f, demonstrating a good GaS stability and reproducibility over several cycles; the sharp rising edge indicates an instant response. Noteworthy, the response time is less than 100 ms (our detection limit), which is two-to-three orders less than other bidimensional semiconductors (for comparison, a response time of MoS_2_ for UV light of ≈41 s has been reported) [20].

The *PDCR* relates to the optical and structural properties, as summarized in Figure 5. The *PDCR* increases with the thickness, with the extinction coefficient, and with the crystal quality of the samples. Specifically, the *PDCR* for the 1665 nm thick sample, calculated at 2 V bias voltage, is 6.7·10^3^ and 3.7·10^5^ under visible and UV light, respectively, whereas it is 40 (UV) and 3.5 (visible) for the 38 nm thin sample.

## 4. Conclusions

In summary, our results provided insights into the interplay between structure, defects, optical properties, and UV photoconductivity of centimeter-size exfoliated GaS samples of various thicknesses. Specifically, we showed that the GaS photoresponsivity increases with increases in the GaS thickness of exfoliated samples. GaS with a thickness >1000 nm results in a UV photoresponsivity of 1.5·10^−4^ AW^−1^ with a photocurrent *on/off* ratio of ~10^5^ stable over several *on/off* cycles and a quick response time of less than 100 ms. As the thickness of the exfoliated GaS samples decreases with increases in exfoliation runs, the photoresponsivity also decreases due to defects introduced by the exfoliation process and a lower light absorption.

These results are useful in designing a new generation of layered GaS UV photodetectors for medical diagnostics, ozone sensing, and optical communication systems.

## Figures and Tables

**Figure 1 nanomaterials-12-00465-f001:**
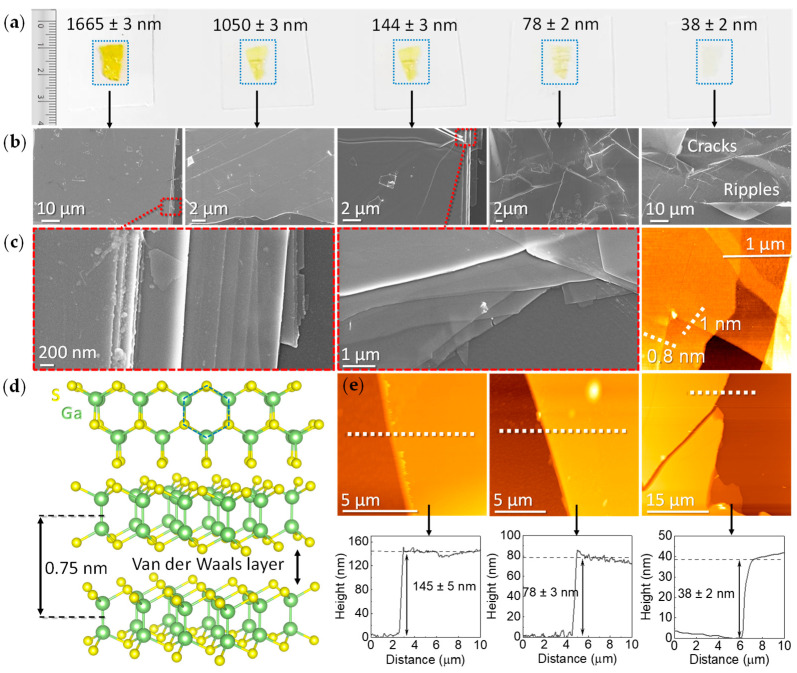
(**a**) Picture of exfoliated GaS of various thicknesses transferred onto glass. (**b**) Corresponding SEM images of the exfoliated GaS samples in (**a**) as a function of thickness. (**c**) Details from SEM and AFM images of the edges of samples to show the GaS layered structure. (**d**) Top view and side view of the GaS crystal structure, showing the monolayer composed of S–Ga–Ga–S bonds and the Van der Waals interaction between layers; the thickness of 0.75 nm for a monolayer is also shown. (**e**) AFM images and line profiles for the 144, 78, and 38 nm samples, from which average thickness has been determined.

**Figure 2 nanomaterials-12-00465-f002:**
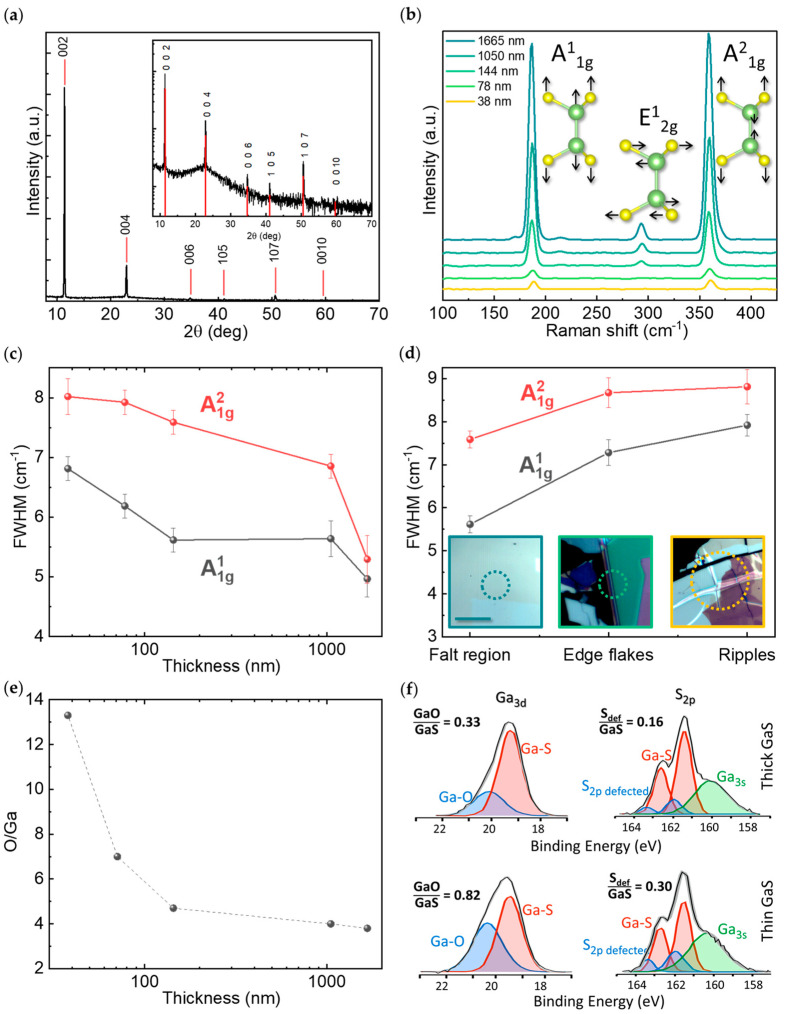
(**a**) XRD diagram of the 38 nm exfoliated GaS. (**b**) Raman spectra of exfoliated GaS with different thicknesses. (**c**) Dependence of the FWHM of the A^1^_1g_ and A^2^_1g_ modes as a function of the GaS average thickness. (**d**) FWHM of the A^1^_1g_ and A^2^_1g_ modes measured at the flat basal plane of the flake, at the edge of the flakes, and at the ripple position (as shown by the micrographs in the inset). (**e**) O/Ga atomic percentage ratio as determined from the Ga3d and O1s XPS peaks as a function of flakes thickness. (**f**) XPS spectra of the Ga3d and S2p photoelectron core levels for the thin and thick GaS flakes. The ratio of the Ga-oxide (GaO) to GaS fitting components from the Ga3d and that of the defective sulfur to GaS from the S2p are also indicated.

**Figure 3 nanomaterials-12-00465-f003:**
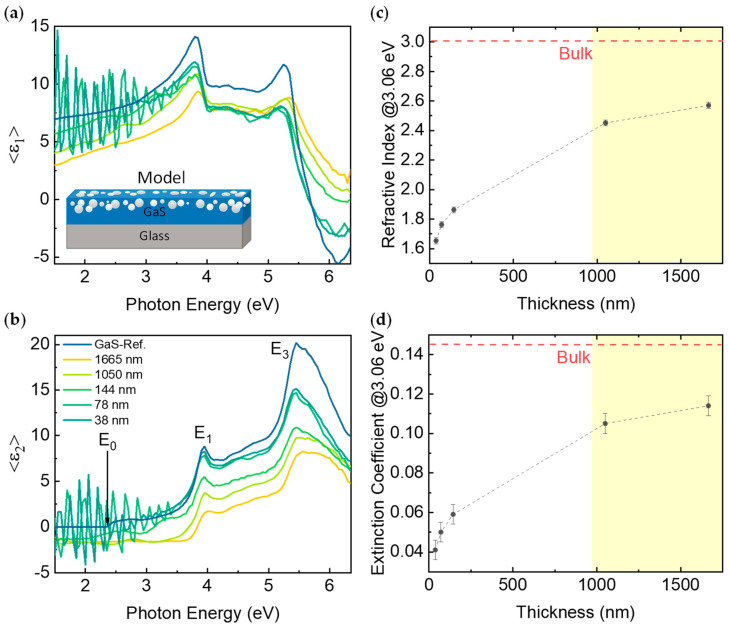
(**a**) Real ⟨ε_1_⟩, and (**b**) imaginary, ⟨ε_2_⟩, parts of the pseudodielectric function, ⟨ε⟩ = ⟨ε_1_⟩ + *i*⟨ε_2_⟩, of exfoliated GaS with different thicknesses. (**c**) Extinction coefficient and (**d**) refractive index at 3.06 eV as a function of thickness. The reference values for the refractive index of bulk crystal GaS are also shown for comparison. The error bar on the refractive index and extinction coefficient comes from the errors on the GaS volume fraction percentage and voids volume fraction percentage of the BEMA approximation of the ellipsometric fitting.

**Figure 4 nanomaterials-12-00465-f004:**
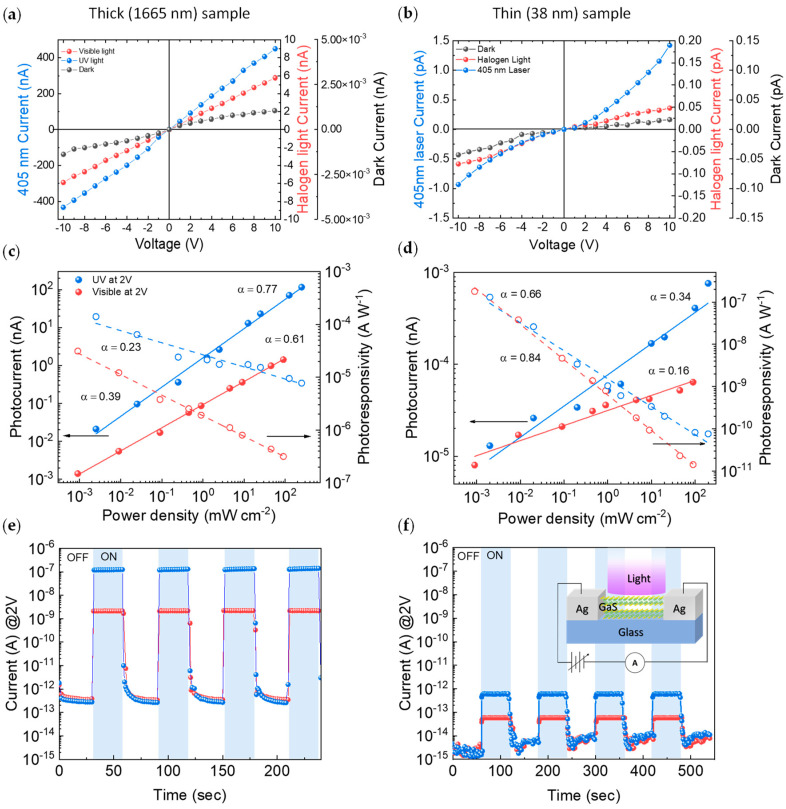
I–V characteristic of (**a**) thick and (**b**) thin exfoliated GaS samples in dark, visible light (100 mW cm^−2^), and UV-light (250 mW cm^−2^) irradiation. Photocurrent and photoresponsivity at 2 V as a function of visible and UV light power density for the (**c**) thick and (**d**) thin GaS samples; photocurrent versus time in response to light *on/off* under visible and UV light (the voltage is kept constant at 2 V) for (**e**) thick and (**f**) thin GaS samples.

**Figure 5 nanomaterials-12-00465-f005:**
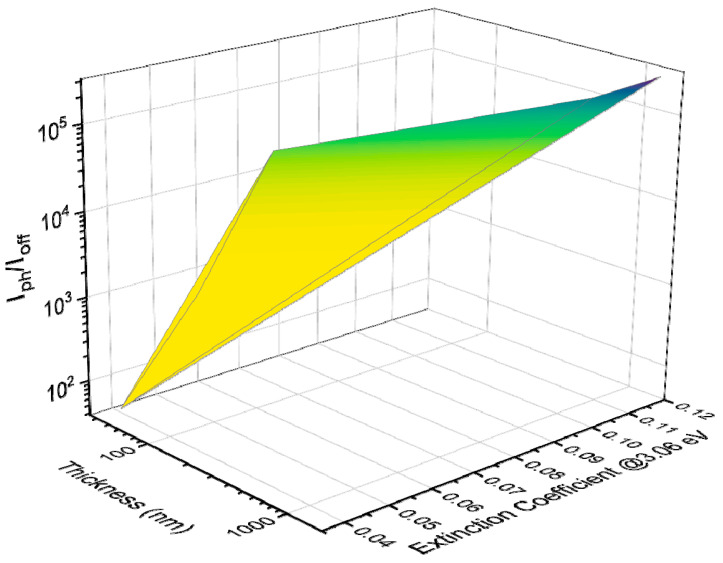
Photo-to-dark current ratio *I_ph_/I_off_* under UV-light (405 nm) irradiation as a function of the thickness and of the extinction coefficient at 3.06 eV.

**Table 1 nanomaterials-12-00465-t001:** Responsivity in typical low-dimensional photodetectors with experimental conditions.

Material	Responsivity (A·W^−1^)	Voltage (V)	Detection Range Wavelength (nm)	Reference
GaS (on glass)	3.1·10^−5^	2	Visible	This work
GaS (on glass)	1.4·10^−4^	2	405	This work
GaS (on SiO_2_/Si)	2.3·10^−4^	2	490	[14]
GaS (on SiO_2_/Si)	4.7	20	275	[22]
GaS (on SiO_2_/Si)	1.9	20	275	[32]
Graphene	5·10^−4^		1550	[33]
Graphene	8.61		Visible-MIR	[34]
MoS_2_ (on SiO_2_/Si)	5·10^−4^	1	Visible	[35]
SnS_2_ (on SiO_2_/Si)	8.8·10^−3^	2	457	[36]

## Data Availability

The data in this study are available from the corresponding author upon request.

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
