# Peer review of "Interplay between Thickness, Defects, Optical Properties, and Photoconductivity at the Centimeter Scale in Layered GaS"

_nanomaterials, 2022, doi:10.3390/nano12030465_

Round 1

Reviewer 1 Report

The authors reported the controlled GaS samples’ thickness through mechanical exfoliation. They had systemically studied the properties of structural, morphological, optical and photoresponsivity as a function of thickness had been systemically studied. It is shown that the photoresponsivity increases with the increase in the GaS thickness. The results are helpful to design GaS UV photodetectors.

This work is well written and organized. The results are clearly presented. The methods are adequately described and are sufficiently detailed to be understood. The results support the conclusions. I would suggest the authors discuss the error bars in their experimental results carefully, for example, the errors in Fig. 1(e) and Fig. 3(d). I recommend the publication of the paper in Nanomaterials with a minor revision.

Author Response

We are very grateful to the reviewer for the positive view of the manuscript.

We also thank the reviewer for the appropriate comment about the errors. A sentences has been added at pg. 3 (lines 111-116) to comment on errors of Fig. 1(e).

We have estimated errors for the thickness from AFM also on the basis of statistical analysis of 10 line profiles in random points of each sample and errors are now provided in Fig. 1e.

We have also added error bars in Fig. 3d and commented the errors in the caption.

Reviewer 2 Report

  1. Authors should pay attention to the sentence structures and typos throughout the manuscript. For example, in introduction (line 31) “enrgy" should be “energy”.
  2. In line 34 and 35, authors have written the following:

“…… whereas bandgap values of 3.02 eV [11] and 3.33 eV [12] have been reported for bilayer and monolayer GaS, respectively …..”

Are authors referring to direct or indirect band gap?

  1. Line 45: Authors have mentioned the following:

“ ……….. well as in very small flakes with controlled thickness ………”

In this line, are authors referring to controlled or uncontrolled thickness? If it is producing samples with controlled thickness, then what is importance of this study?

  1. Line 60, authors should mention manufacturer/supplier of Gallium (II) Sulfide (GaS) crystal.
  2. Line 260, “… various thickness ….” should be “…various thicknesses..”.
  3. Line 100 to 102, the meaning is not clear. Does the band gap change with thickness? Is it the direct or indirect band gap?

Author Response

Authors: we acknowledge the reviewer for the helpful suggestions to improve the manuscript using the provided comments. Specifically:

  1. Authors should pay attention to the sentence structures and typos throughout the manuscript. For example, in introduction (line 31) “enrgy" should be “energy”.

Authors: The manuscript has been carefully checked for typos.

  1. In line 34 and 35, authors have written the following:

“…… whereas bandgap values of 3.02 eV [11] and 3.33 eV [12] have been reported for bilayer and monolayer GaS, respectively …..”

Are authors referring to direct or indirect band gap?

Authors: we have specified that we refer to the indirect bandgap.

  1. Line 45: Authors have mentioned the following:

“ ……….. well as in very small flakes with controlled thickness ………”

In this line, are authors referring to controlled or uncontrolled thickness? If it is producing samples with controlled thickness, then what is importance of this study?

Authors: we have rewritten the sentence, which was not well written and misleading; we refer to uncontrolled thickness.

  1. Line 60, authors should mention manufacturer/supplier of Gallium (II) Sulfide (GaS) crystal.

Authors: The manufacturer has been specified at pg. 2 (line 60).

  1. Line 260, “… various thickness ….” should be “…various thicknesses..”.

Authors: corrected.

  1. Line 100 to 102, the meaning is not clear. Does the band gap change with thickness? Is it the direct or indirect band gap?

Authors: we have rewritten the sentence, and specified that the bandgap does not change in the investigated thickness range.

Reviewer 3 Report

In this work GaS samples of various thickness in the range from 30 to 1200 nm have been obtained by mechanical exfoliation to study the interplay between structural, morphological, optical and photoresponsivity properties as a function of thickness.

The samples were characterized using by Raman spectroscopy and XRD, the morphology by SEM & AFM, density and optical properties by spectroscopic ellipsometry, and photoresponsivity by current-voltage measurements under UV-light.

This is an interesting work; Nevertheless some revisions are needed in order to publish the specific work.

  1. XRD is not actually a spectrum. Please revise the term "spectrum" to "diagram".
  2. The authors state that there is no significant shift in the frequency of both the A11g and A21g Raman modes, contrarily to what typically observed for other bidimensional chalcogenides. Could they discuss a little bit more their findings?
  3. A few typos should be checked.

Author Response

We gratefully acknowledge the reviewer for the positive comments and suggestions. We have made changes accordingly, as specified point-by-point:

  1. XRD is not actually a spectrum. Please revise the term "spectrum" to "diagram".

Authors: corrected according to suggestion.

  1. The authors state that there is no significant shift in the frequency of both the A11g and A21g Raman modes, contrarily to what typically observed for other bidimensional chalcogenides. Could they discuss a little bit more their findings?

Authors: a new sentence has been added at pg. 5 (line 152-164) to discuss the Raman peaks position.

  1. A few typos should be checked.

Authors: typos have been corrected.